# A Tracking Imaging Control Method for Dual-FSM 3D GISC LiDAR

Yu Cao [1,2,3,4,*], Xiuqin Su [1], Xueming Qian [2], Haitao Wang [1], Wei Hao [1], Meilin Xie [1], Xubin Feng [1], Junfeng Han [1], Mingliang Chen [5] and Chenglong Wang [5]

1 Xi'an Institute of Optics and Precision Mechanics of CAS, Xi'an 710119, China; suxiuqin@opt.ac.cn (X.S.); wanghaitao@opt.ac.cn (H.W.); haowei@opt.ac.cn (W.H.); xiemeilin@opt.ac.cn (M.X.); fengxubin@opt.ac.cn (X.F.); hanjf@opt.ac.cn (J.H.)
2 School of Electronic and Information Engineering, Xi'an Jiaotong University, Xi'an 710049, China; qianxm@mail.xjtu.edu.cn
3 University of Chinese Academy of Sciences, Beijing 100049, China
4 CAS Key Laboratory of Space Precision Measurement Technology, Xi'an 710119, China
5 Key Laboratory for Quantum Optics and Center for Cold Atom Physics of CAS, Shanghai Institute of Optics and Fine Mechanics, Chinese Academy of Sciences, Shanghai 201800, China; chenmingliang@siom.ac.cn (M.C.); wangchenglong@siom.ac.cn (C.W.)
* Correspondence: caoyu@opt.ac.cn; Tel.: +86-132-0180-8819

**Abstract:** In this paper, a tracking and pointing control system with dual-FSM (fast steering mirror) composite axis is proposed. It is applied to the target-tracking accuracy control in a 3D GISC LiDAR (three-dimensional ghost imaging LiDAR via sparsity constraint) system. The tracking and pointing imaging control system of the dual-FSM 3D GISC LiDAR proposed in this paper is a staring imaging method with multiple measurements, which mainly solves the problem of high-resolution remote-sensing imaging of high-speed moving targets when the technology is transformed into practical applications. In the research of this control system, firstly, we propose a method that combines motion decoupling and sensor decoupling to solve the mechanical coupling problem caused by the noncoaxial sensor installation of the FSM. Secondly, we suppress the inherent mechanical resonance of the FSM in the control system. Thirdly, we propose the optical path design of a dual-FSM 3D GISC LiDAR tracking imaging system to solve the problem of receiving aperture constraint. Finally, after sufficient experimental verification, our method is shown to successfully reduce the coupling from 7% to 0.6%, and the precision tracking bandwidth reaches 300 Hz. Moreover, when the distance between the GISC system and the target is 2.74 km and the target flight speed is 7 m/s, the tracking accuracy of the system is improved from 15.7 µrad (σ) to 2.2 µrad (σ), and at the same time, the system recognizes the target contour clearly. Our research is valuable to put the GISC technology into practical applications.

**Keywords:** 3D GISC LiDAR; dual-FSM tracking and aiming; FSM feedback decoupling; composite axis control; remote-sensing imaging

## 1. Introduction

Three-dimensional GISC LiDAR based on sparse and redundant representation is a new LiDAR imaging system that combines the spatial fluctuation characteristics of light fields and modern information theory. Its imaging field of view is independent of resolution. The wide-field staring imaging mode can be used to capture moving targets for high-resolution imaging detection [1–3]. Compared with the flash camera LiDAR, which needs to distribute the reflected light signal of the target on the focal plane array photodetector, the 3D GISC Lidar only needs a single pixel detector with no spatial resolution to receive all the reflected light signal of the target scene. Therefore, the imaging detection sensitivity of the system can be greatly improved. In addition, in the process of imaging detection,

3D GISC LiDAR can make full use of various prior constraints of the image, so as to break through the requirements of the Nyquist sampling theorem for sampling times and greatly improve the efficiency of image information acquisition [4]. Recently, 3D GISC LiDAR has completed a proof-of-principle demonstration experiment under a real atmospheric environment, which showed its advantages in target imaging detection. Because 3D GISC LiDAR is a multimeasurement staring imaging method, the data acquisition time is relatively long and is not applicable for real-time imaging. The relative motion of the target and the system leads to a decrease in image resolution [5]. In order to solve the above problems, the team at the Shanghai Institute of Optics and Fine Mechanics Chinese Academy of Sciences proposed a moving target imaging radar system based on a single FSM tracking [4], and successfully realized an airborne flight test with a spatial resolution and range resolution better than 0.5 m at a flight altitude >1 km. However, this method has the problem that the system receives aperture constraints, and due to the tracking accuracy defect of an FSM, there is still the problem of motion blur in moving-target tracking [6,7].

The photoelectric tracking system is one of the important pieces of equipment to achieve the functions of capturing, tracking, and aiming. It plays a pivotal role in the fields of target detection, navigation and positioning, fire control systems, aerospace, telescope systems, beam stabilization control, and space optical communications [8]. Recently, more and more requirements have been put forward for the stability and tracking accuracy of photoelectric tracking systems. In the field of quantum correlation imaging, due to the influence of the system bandwidth, moment of inertia, and tracking range, it is very challenging to only use a large inertia tracking mechanism. However, a small inertia tracking system has difficulties meeting the requirements of the system working range. Therefore, large inertia tracking systems are usually designed to work together with small inertia tracking systems to form a composite axis control system to meet the requirements of tracking accuracy [9,10].

The tracking accuracy improvement method of a compound axis control system is mainly divided into two aspects: coarse tracking and fine tracking.

For coarse tracking, staring in the 1950s, researchers began to conduct in-depth research on coarse tracking control technology from the most basic PID (proportional–integral–derivative) control algorithm [10]. After decades of development, the research methods gradually expanded to advanced control algorithms, such as fuzzy PID control, adaptive control, optimal control, variable structure control, etc. The technology of improving the coarse tracking accuracy of the system is very mature [10].

For fine tracking, at present, the fine tracking actuator of the composite axis tracking and aiming system is a high-speed mirror deflection mechanism, which is also known as fast steering mirror (FSM). It mainly consists of a piezoelectric ceramic drive and a VCA (voice coil actuator) drive [11]. The FSM driven by piezoelectric ceramics has the advantages of a high precision and fast response, but the driving range is only tens of microns. The FSM driven by a VCA has the advantages of a high precision, large travel range, fast response speed, and low driving voltage. It is widely used to suppress beam jitter with large values. The FSM's output accuracy determines the control accuracy of the beam dithering system [12–15]. However, when the FSM rotates in response to two free degrees, there will be mechanical coupling, that is, the response of one degree of freedom will cause the response of the other free degree. On the one hand, the coupling of the motor's position feedback information is caused by the noncoaxial installation of motor and sensor, On the other hand, the X-axis and Y-axis of the mirror motion are coupled due to the installation error of the motor and the mirror. The existing method is to calculate the displacement of the X and Y axes of the FSM through the coordinate solution method according to the installation angle between the sensor and the motor. However, the existing solutions usually ignore the installation deviation of the included angle and the deviation generated in the process of motor movement.

In this paper, in order to further solve the constraint problem of the system receiving aperture and the motion blur problem of the tracking mode of the single pendulum mirror,

we undertook a joint research project with the Shanghai Institute of Optics and Fine Mechanics Chinese Academy of Sciences, and propose a dual-FSM tracking and aiming control method based on 3D GISC LiDAR. Firstly, we propose a decoupling method combining motion decoupling and sensor decoupling for the motor's position feedback information coupling and the mirror motion coupling. Through this method, the deflection response of the FSM in each degree of freedom can be decoupled, and the feedback information of each degree of freedom can also be decoupled. Secondly, we propose a method to suppress the inherent mechanical resonance of the FSM by adding a digital notch filter to the position loop. Finally, based on the imaging principle of GISC and the constraint of system receiving aperture, we propose an optical path design diagram of tracking imaging system based on dual-FSM 3D GISC LiDAR. This method greatly improves the detection ability of 3D GISC Lidar to high-speed moving targets. Therefore, this research is of great significance for the achievement transformation of 3D GISC Lidar technology for practical application.

The rest of this paper is organized as follows: in Section 2, the solution principle of the feedback position information of the FSM and the main factors affecting its accuracy are introduced; In Section 3, the digital decoupling method for the motor's position feedback information coupling and mirror motion coupling is introduced in detail. In addition, the first-order resonance suppression method of the FSM is also introduced. In Section 4, the optical path design method of the dual-FSM tracking imaging system based on 3D GISC LiDAR is introduced. Section 5 introduces the tracking accuracy experiment and imaging experiment of the dual-FSM tracking imaging system based on 3D GISC LiDAR. In Section 6, conclusions are given.

## 2. Related Work

Because an FSM has the characteristics of a small inertia, fast response, high bandwidth and large acceleration, it is generally used for LOS (line-of-sight) stabilization during sensor exposure or fast compensation of image motion in swing scanning of large-inertia frames. In the field of quantum correlation imaging, FSM has two actuation modes: piezoelectric ceramic and VCA. Compared with piezoelectric ceramic actuation, voice coil actuation of an FSM has the characteristics of a simple structure, low driving voltage, small volume, and large stroke.

At present, the commonly used FSMs are composed of a mirror, a VCA, displacement sensor, and controller. The eddy-current displacement sensor is the measurement feedback link of the fast reflector, which detects the displacement change of the VCA of the galvanometer and transmits it to the controller in real time. According to coordinate transformation, the controller calculates the displacement driven by the VCA on the actual X/Y axis. After decoupling, the controller outputs the control quantity to the VCA-running mechanism according to the control algorithm, and the whole system forms a real-time closed-loop control system [15].

Chang et al. proposed a novel FSM compensation system, which made the existing compensation system have the advantages of a shorter optical path length, fewer components, and an easier way to be set in different positions [16]. Sun et al. designed an FSM flexible support structure based on four cross-axis flexural hinges, and used eddy-current sensors to indirectly collect the motor's position information, but did not mention a decoupling method of measurement information [17]. Huang et al. proposed a reduced order ADRC (active disturbance rejection control) method based on an FSM actuated by a VCA [18], which can improve the dynamic performance of FSM. However, it regarded the results of the eddy-current sensor as known, so it did not consider the decoupling strategy of the position feedback information [18]. Wang et al. proposed a dual-feedforward and dual-neural-network adaptive decoupling control algorithm based on an analog control FSM, which compensated dc coupling component and non-dc coupling component, respectively, and the coupling degree was reduced from about 5% to less than 1.0 ‰. However, because the FSM mentioned in that paper adopted a full analog circuit control, considering

the application environment usually required by an FSM, on the one hand, the position feedback signal acquisition and processing anti-interference ability of the FSM was weak, resulting in a low control accuracy. On the other hand, compared with the scheme of a full digital control of a full analog circuit, it was more difficult to expand to applications [19].

Compared with existing approaches, the FSM driven by a VCA designed in this paper is shown in Figure 1. This structure can realize the deflection of two degrees of freedom on the X-axis and Y-axis. We installed two VCAs on each axis at a distance R from the center point O. The position measurement was realized by installing eddy-current sensors at the positions of the X'-axis and the Y'-axis at a distance R from the center point O. The included angle between the X'-axis and X-axis was 45 degrees, the included angle between the Y'-axis and Y-axis was 45 degrees, and the X-axis and Y-axis and the X'-axis and Y'-axis were at right angles to each other [20–23].

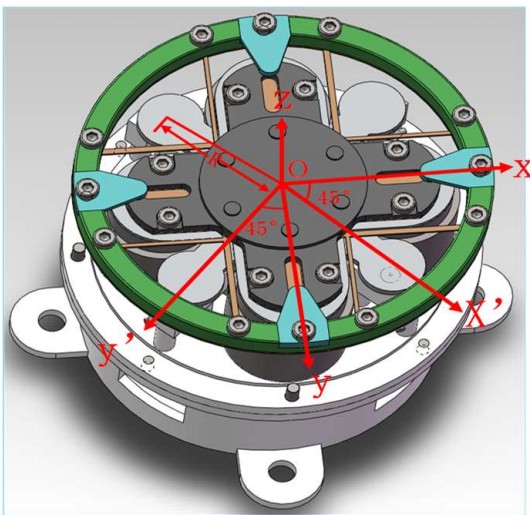

**Figure 1.** Structure of a fast steering mirror.

When the VCA on the X-axis and Y-axis moves at a certain angle, the measured values of the eddy-current sensors on the X'-axis and Y'-axis also change correspondingly, and its coordinate system is shown in Figure 2.

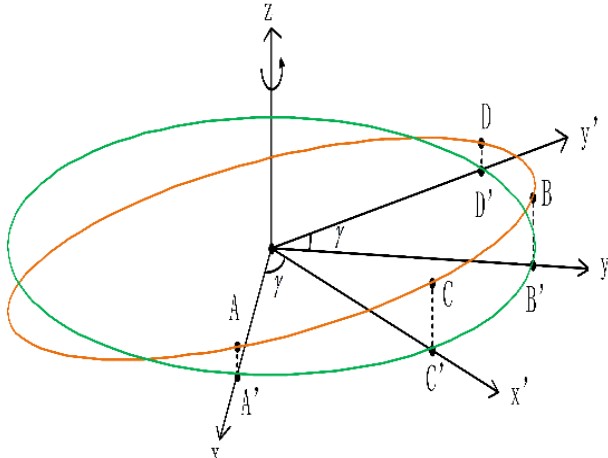

**Figure 2.** Sensor Coordinates.

As shown in Figure 2, the FSM position coordinate system takes the Z-axis in the rectangular coordinate as the rotation axis, and the plane *xoy* is *x′oy′* after rotating $\gamma$ around the Z-axis. Through the right-hand helix theorem, the right-hand helix direction is the positive direction of object rotation, that is, it is counterclockwise from the origin

of the positive half axis of the Z-axis. As shown in Figure 2, the red curve is the intersection of the plane AOB and cylinder $x^2 + y^2 = R^2$, where the coordinates of points A and B are $A(R, 0, Z_A)$, $B(0, R, Z_B)$, respectively, then the equation of the plane AOB is: $Z_A x + Z_B y - Rz = 0$. As shown by the blue line in Figure 2, since A′, B′, C, D, and O are in one plane, the projection of the elliptic ABCD curve on $xoy$ is the circle A′B′C′D′, and the projections of A, B, C, and D on the plane are points A′, B′, C′, and D′, respectively, where C′, and D′ are points on $ox'$ and $oy'$, and the coordinates of C′ and D′ are $(R \cos \gamma, R \sin \gamma, 0)$ and $(-R \sin \gamma, R \cos \gamma, 0)$, respectively, so we can set the coordinates of C and D as $(R \cos \gamma, R \sin \gamma, Z_C)$ and $(-R \sin \gamma, R \cos \gamma, Z_D)$, respectively [24,25].

Because A, B, C, D, and O are on the same plane, the coordinates of points C and D can be substituted into the equation of the plane AOB, respectively:

$$\begin{cases} R \cos \gamma Z_A + R \sin \gamma Z_B - R Z_C = 0 \\ -R \sin \gamma Z_A + R \cos \gamma Z_B - R Z_D = 0 \end{cases} \quad (1)$$

And we have:

$$\begin{cases} \cos \gamma Z_A + \sin \gamma Z_B - Z_C = 0 \\ -\sin \gamma Z_A + \cos \gamma Z_B - Z_D = 0 \end{cases} \quad (2)$$

Considering that the motor and sensor are ideally installed at an angle of 45 degrees, therefore, when $\gamma = 45°$ is brought into Equation (2), it becomes:

$$\begin{cases} \frac{\sqrt{2}}{2}(Z_A + Z_B) = Z_C \\ \frac{\sqrt{2}}{2}(-Z_A + Z_B) = Z_D \end{cases} \quad (3)$$

Solving Formula (3), we have:

$$\begin{cases} Z_A = \frac{\sqrt{2}}{2}(Z_C - Z_D) \\ Z_B = \frac{\sqrt{2}}{2}(Z_C + Z_D) \end{cases} \quad (4)$$

Formula (4) shows that in the ideal state and without considering the installation deviation between the motor and the mirror, since the vertical coordinate values of C and D are known and points A, B, C, D, and O are on one plane, the vertical coordinate values of points A and B can be obtained through the equation.

In practical applications, when considering the installation deviation between the motor and the sensor, the relationship between the displacement of the VCA on the X-axis and Y-axis and the measured value of the 45-degree-inclined eddy-current sensor is shown in Equations (5) and (6), where $S_a$ and $S_b$ are the displacements output by the eddy current sensor, $\theta_1$ is the installation deviation between the eddy-current sensor and the X-axis, and $\theta_2$ is the installation deviation between the eddy-current sensor and the Y-axis.

$$\Delta X = S_a * \cos\left(45° + \theta_1\right) - S_b * \sin\left(45° + \theta_1\right) \quad (5)$$

$$\Delta Y = S_a * \sin\left(45° + \theta_2\right) + S_b * \cos\left(45° + \theta_2\right) \quad (6)$$

## 3. Methods for Digital FSM Decoupling Control and Resonance Suppression Algorithm

In this section, by studying the existing FSM control algorithm, a digital decoupling method is proposed for the motor's position feedback information coupling and mirror motion coupling not considered in the solution process. Aiming at the low-frequency first-order resonance of the FSM caused by a flexure hinge as a support structure, we propose a suppression method of the FSM's first-order resonance. The digital decoupling method consists of two parts: motion decoupling and sensor decoupling. Motion decoupling is used to solve the X-axis and Y-axis motion coupling of the mirror caused by the installation error of the motor and the mirror, and sensor decoupling is used to solve the motor's

position feedback information coupling caused by the noncoaxial installation of the motor and the sensor.

In this paper, we propose the schematic diagram of a single FSM high-precision control system. It is shown in Figure 3. It includes motion decoupling, sensor decoupling, position controller, notch filter, current controller, H-bridge drive, PWM (pulse-width modulation) generation, current sampling, and other links for the X/Y axis of a single FSM.

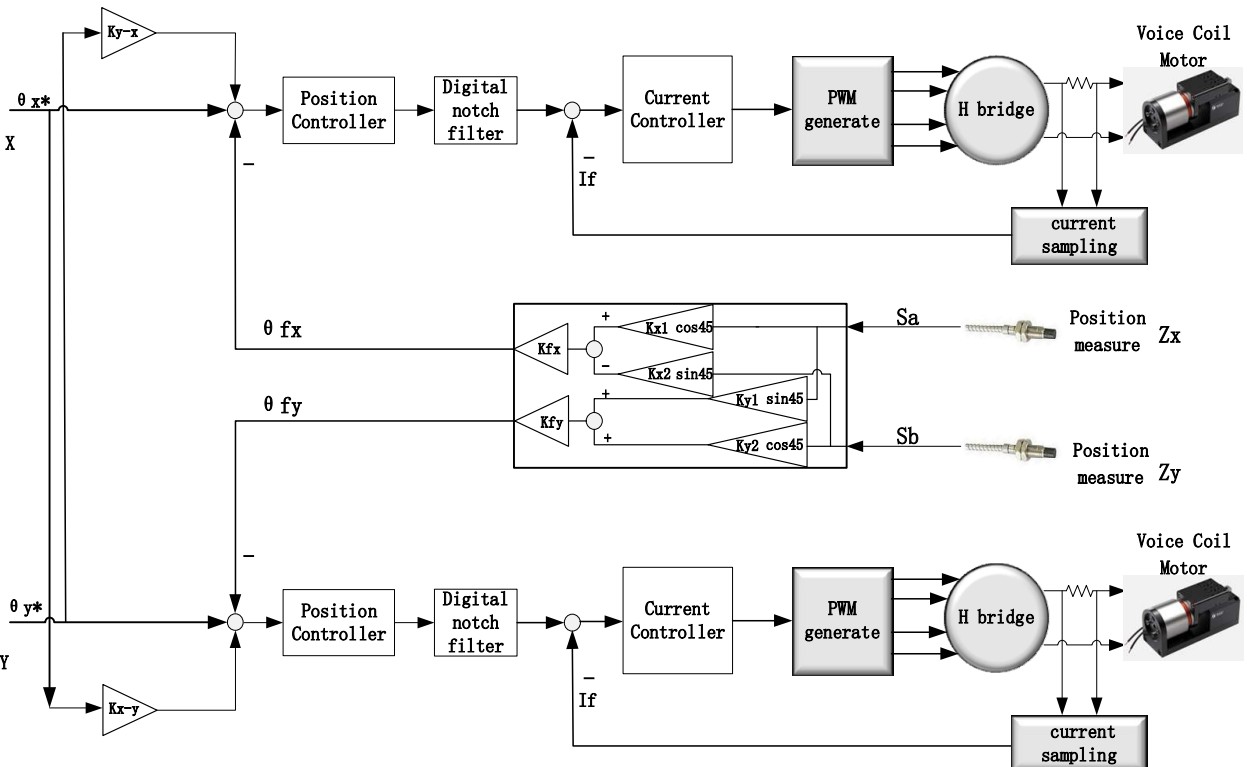

**Figure 3.** FSM control principle diagram.

### 3.1. Motion Decoupling

Firstly, the sensor part is peeled off. There is inherent coupling between X-axis and Y-axis when they are in motion, and the coupling size has a linear relationship with the motion deflection angle. Take the movement of the X-axis as an example. When the position loop is open, the X-axis actuator is given a fixed current, and the Y-axis movement state is monitored through the light pipe. $K_{x-y}$ is obtained as shown in Equation (7) below. As shown in Equation (8) below, $K_{y-x}$ can also be obtained in the same way.

$$\Delta Y = K_{x-y} * \Delta X \tag{7}$$

$$\Delta X = K_{y-x} * \Delta Y \tag{8}$$

Then, by adjusting $K_{x-y}$ and $K_{y-x}$, the corresponding relationship between the given position and the output position is adjusted to realize the motion decoupling of the X and Y axes.

### 3.2. Sensor Decoupling

After the motion decoupling, the motion can be considered independent between the X and Y axes. Then, the sensor error is further corrected. We present a method of sensor-decoupling correction in this paper, as shown in Figure 3, where $K_{fx}$ and $K_{fy}$ are feedback coefficients. Taking the X-axis as an example, first, we make the X-axis move

independently in an open loop, and the Y-axis is stationary, and then, we get the feedback function relationship between the Y-axis and the X-axis.

$$\Delta X = K_{y-x} * \Delta Y + K_{fy}(K_{x1} * S_a * \cos(45°) + K_{x2} * S_b * \sin(45°)) \qquad (9)$$

$$\Delta Y = K_{x-y} * \Delta X + K_{fx}(K_{y1} * S_a * \sin(45°) + K_{y2} * S_b * \cos(45°)) \qquad (10)$$

The calculation of the decoupled Y-axis feedback position information is shown in Formula (10). By adjusting $K_{y1}$, $K_{y2}$, and $K_{fx}$, the Y-axis feedback becomes zero, and we use this method to set the parameters in the calculation formula. The decoupled X-axis feedback position information is calculated as shown in Formula (9). Similarly, the parameters in the formula can also be adjusted according to the methods mentioned above. First, the Y-axis moves independently in an open loop, the X-axis is stationary, and $K_{x1}$, $K_{x2}$, and $K_{fy}$ are adjusted to make the X-axis feedback go to zero. Finally, the sensor decoupling is realized by this calculation method.

### 3.3. Research on Suppression Method of First-Order Resonant Frequency of FSM

Because the first-order resonant frequency of the FSM driven by a VCA is low, mostly 30–60 Hz, which is far lower than the fine-tracking bandwidth of an FSM, in order to suppress the inherent first-order mechanical resonance, a suppression method consisting of a position loop and a second-order digital notch filter is proposed in this paper. The first-order resonant frequency of the system can be measured through the frequency sweep test of the inner loop of the system. Taking the first-order resonant frequency of 44 Hz as an example, a second-order notch filter with a frequency of 44 Hz can be designed at the output of the position ring.

$$G(s) = \frac{1 + S^2 R^2 C^2}{1 + SRC/Q + S^2 R^2 C^2} \qquad (11)$$

Among them, $RC = \frac{1}{\omega} = \frac{1}{2\pi f}$, $Q = \frac{\omega_0}{BW} = \frac{44}{3.11}$.

The spectrum characteristics of the second-order filter are shown in the simulation from Figure 4. It can be observed that the attenuation at the notch frequency of 44 Hz is greater than 120 dB. After calculation, the stopband ripple is $4.47 \times 10^{-7}$ and the passband cut-off frequencies are 43 Hz and 45 Hz. Therefore, the control parameters of the servo system can be adjusted to the optimal state under the action of the notch filter.

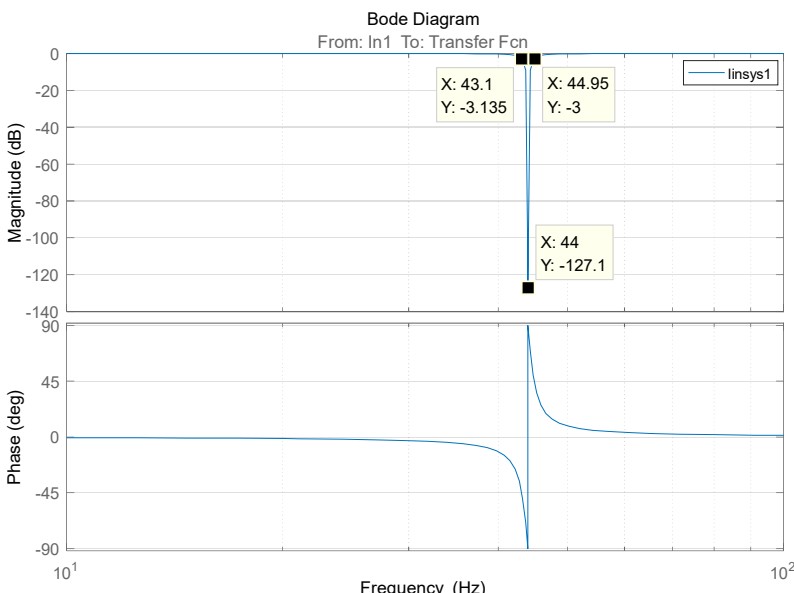

**Figure 4.** Spectrum characteristics of second-order digital filter.

The test results are shown in Figure 5a for the position feedback curve of a low-frequency sinusoidal oscillation of the X-axis. According to the curve, it can be seen that the first-order resonance effect is relatively obvious. After using the position loop and digital notch filter method to suppress the first-order resonance, as shown in Figure 5b, given a sine of 300 Hz and 1 mrad at the X-axis position, the position feedback curve (blue) is shown. We find that the vibration caused by the first-order resonance cannot be seen on the feedback curve.

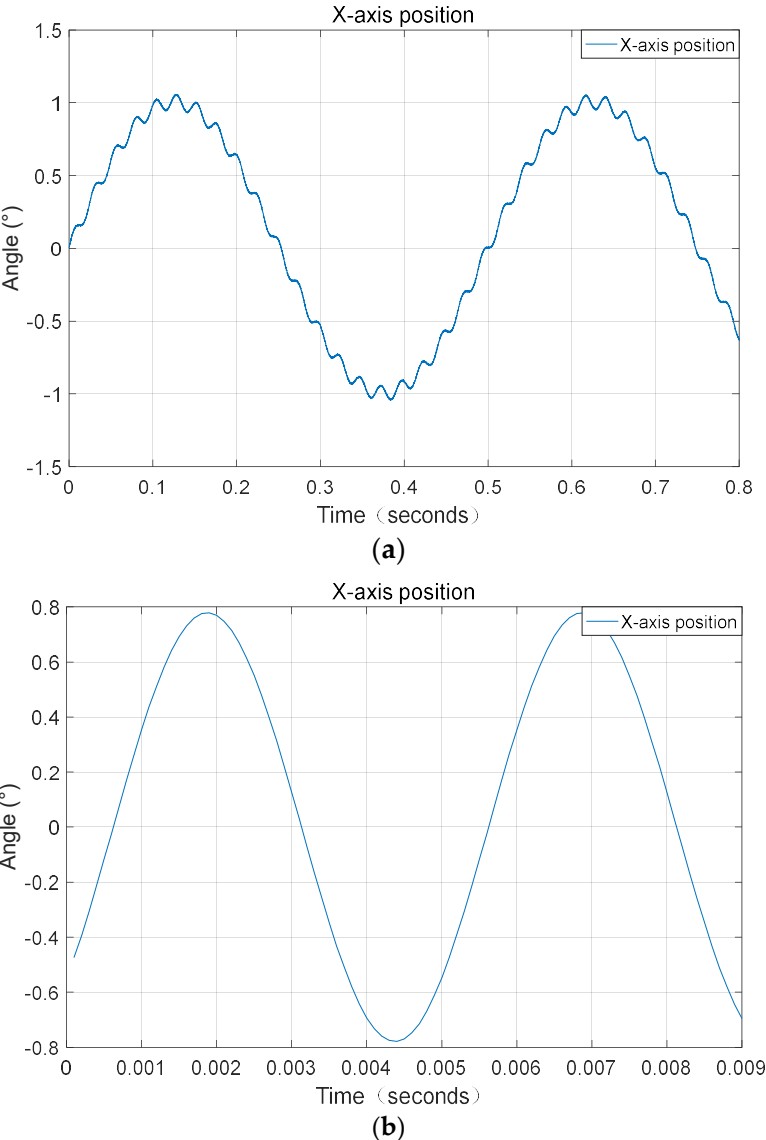

**Figure 5.** Comparison of position curves before and after resonance suppression. (**a**) Before suppressing resonance; (**b**) after suppressing resonance, the X-axis shows a 200 Hz sinusoidal position curve.

The high-precision position tracking controller proposed in this paper is realized by a PI control and digital notch filter. The PI control reduces the tracking error, and the notch filter can solve the resonance problems caused by some frequency components of the control and the first-order mechanical resonance of the structural system.

In this paper, the incomplete integral–proportional controller is introduced into the digital current loop, which can improve the stability and control bandwidth of the circuit. The current loop determines the PWM pulse output to the H-bridge through the duty cycle, which realizes the control of the magnitude and direction of the VCA current, and then controls the magnitude and direction of the output torque. Finally, the four-quadrant operation of the FSM can be realized.

## 4. Design Method of Dual-FSM 3D GISC Tracking Imaging Control System

Three-dimensional GISC LiDAR is a new nonlocal laser-imaging radar mechanism. Different from the traditional "point-to-point" mode of information acquisition, the 3D GISC LiDAR first uses the laser to irradiate the rotating ground glass to generate a speckle field (pseudothermal light field). After passing through the beam splitter, the speckle field is divided into two paths: one is the reference optical path, which records the spatial distribution information of the speckle field intensity with the help of the reference camera, and the other is the object optical path, which projects the speckle field to the target scene to realize the spatial intensity coding of the target, and uses a barrel detector without spatial resolution to record the flight time signal of the target echo. Finally, the three-dimensional information of the target scene is obtained by calculating the second-order correlation between the reference speckle and the target time-of-flight signal [26–28]. From the working principle of a laser three-dimensional intensity correlation imaging radar, it can be seen that on the one hand, the use of spatial intensity coding enables the radar to obtain high-dimensional information of the target with a point detector, which reduces the demand for detection devices, and strengthens its antijamming performance in complex channel environment to a certain extent; on the other hand, the information acquisition mode based on the second-order correlation depends on multiple sampling, so it has an inherent motion-blur problem in the moving target scene [29].

In order to further solve the constraint problem of the system receiving aperture and the motion-blur problem of the tracking mode of the single pendulum mirror, we undertook a joint research project with the Shanghai Institute of Optics and Fine Mechanics Chinese Academy of Sciences, and we propose a scheme of a moving-target laser 3D correlation imaging radar system based on a dual-FSM tracking mode as shown in Figure 6. The system consists of a transmitting system and receiving system, as shown in Figure 6a,b, respectively.

### 4.1. Transmitting System

The transmitting system includes a reference optical path module, an object optical path transmitting module, and an active tracking module. In the transmitting system, the active tracking module uses the spectral band of 480–650 nm for target tracking, and the 420–620 nm photons reflected from the target are imaged on the monitor CCD (charge-coupled device) camera. After the image recorded by the monitoring camera is processed by the tracker, the miss distance is extracted and injected into the controller. Then, the laser three-dimensional correlation real-time imaging is realized by dual-FSM tracking.

The correlation imaging radar device adopts the working wavelength of 1064 nm. Photons are incident on the transmitting FSM in the form of parallel light to realize optical path coupling. Limited by the effective area of the FSM, the transmitting system and receiving system adopt the design scheme of a beam-expanding and collimation method to meet the requirements of imaging resolution and detection signal-to-noise ratio, respectively. A solid-state pulsed laser (with a center wavelength l = 1064 nm, pulsed width of 10 ns and a repetition rate of 2 kHz) illuminates a rotating ground glass and generates speckle field. The speckle field is divided into two paths after passing through the beam splitter. One path is recorded by the local reference camera, and the other path passes through the transmitting mirror and irradiates the target area through the transmitting FSM and the post beam expansion collimation system; at the same time, the active tracking module with a good optical path assembly images the target area with the help of the beam-expanding and collimation system and the transmitting FSM, and realizes the stable tracking of the target through the calculation of the target's miss distance and the transmitting FSM control.

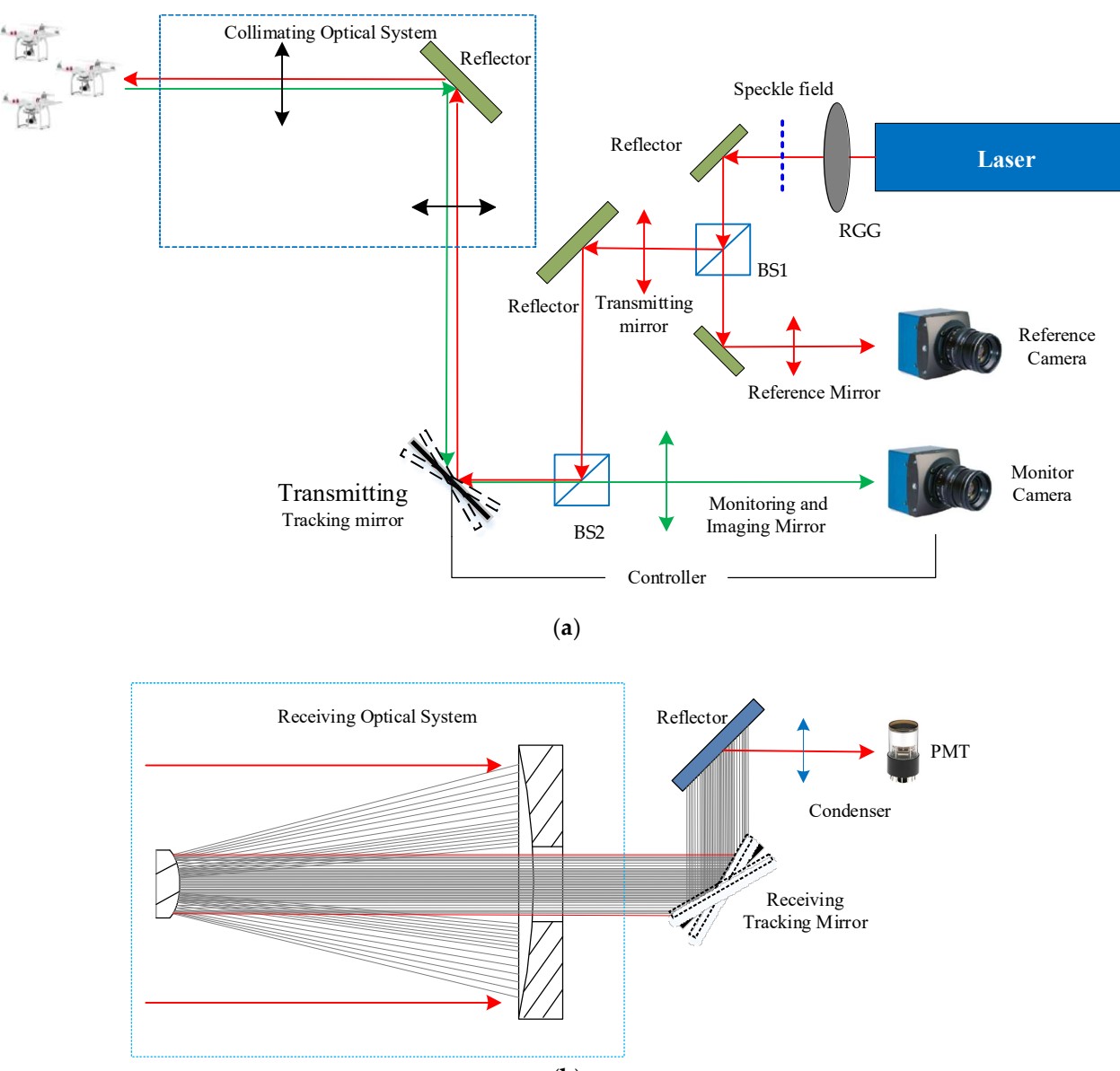

**Figure 6.** Experimental platform's schematic diagram of dual-FSM 3D GISC LiDAR tracking control system (PMT: photon multiplier tube, BS: beam splitter, RGG: rotating ground glass). (**a**) Transmitting system's optical path structure; (**b**) receiving system's optical path structure.

### 4.2. Receiving System

The receiving system includes the object's optical path receiving module and linkage tracking module. In the receiving system, the echo signal from the target is collected by the card-receiving collimator and finally irradiated on the detector (PMT) through the receiving FSM. The FSM deflection angle in the receiving system can be calculated by the active tracking module of the transmitting system according to the optical magnification. Through the cooperation of the dual FSMs of the transceiver system, the stable tracking of moving targets and laser three-dimensional correlation imaging are realized. Its tracking stability determines the strength of the echo signal and at the same time determines the resolution of ghost imaging.

The transmitting tracking mirror is the tracking mirror, and the receiving tracking mirror is the follow-up mirror, and both tracking mirrors are FSMs. After the image recorded by the surveillance camera is processed by the tracker, the missed amount is extracted and injected into the controller. Furthermore, real-time imaging between the 3D

GISC LiDAR and the target is realized through dual-FSM tracking and pointing. Moreover, its tracking stability determines the strength of the echo signal and at the same time determines the resolution of ghost imaging [4].

## 5. Experimental Platform and Result Analysis

Figure 7 shows the photo of the dual-FSM GISC tracking imaging system, and the transmitting FSM in the system is shown in Figure 7a.

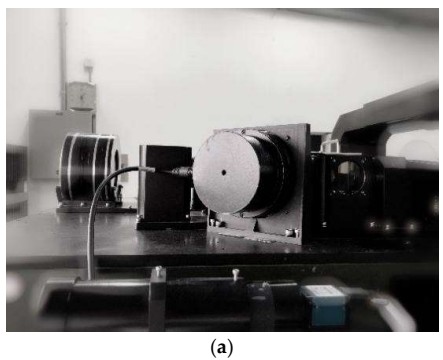 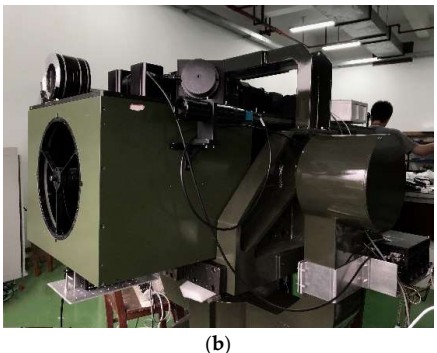

(**a**)                                                                                           (**b**)

**Figure 7.** Photo of the experimental platform. (**a**) Photo of the transmitting FSM; (**b**) dual-FSM 3D GISC LiDAR tracking imaging system.

### 5.1. FSM Decoupling Comparison Test

Taking the X-axis sine swing and Y-axis static as an example, the effect of the sensor decoupling before and after decoupling is shown in Figure 8. We find that the coupling before and after decoupling has dropped from 7% to 0.6%.

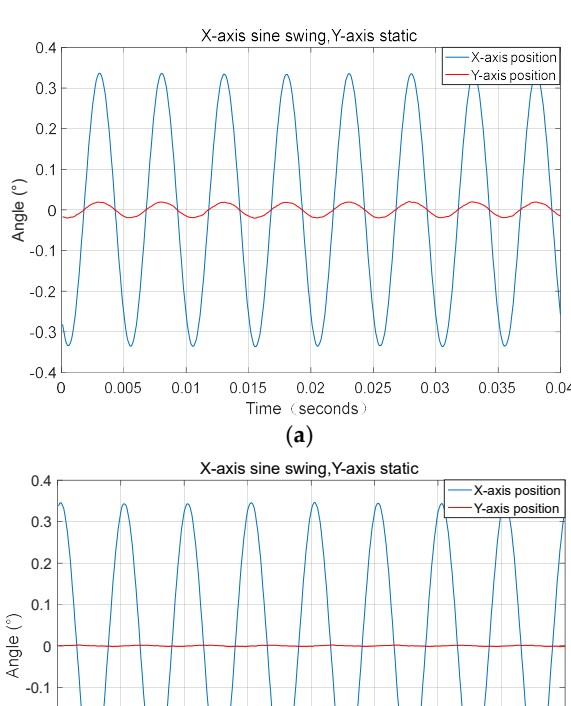

**Figure 8.** Comparison before and after decoupling. (**a**) Before sensor decoupling; (**b**) after sensor decoupling.

### 5.2. FSM Response Bandwidth Test

When the given curve (blue) of the FSM is 300 Hz and the amplitude of the sine is 1 mrad, the position feedback curve (red) is as shown in Figure 9, and it can be seen the X-axis amplitude is 0.72 mrad, and the phase difference is less than 180°, so the X-axis bandwidth of the FSM reaches 300 Hz.

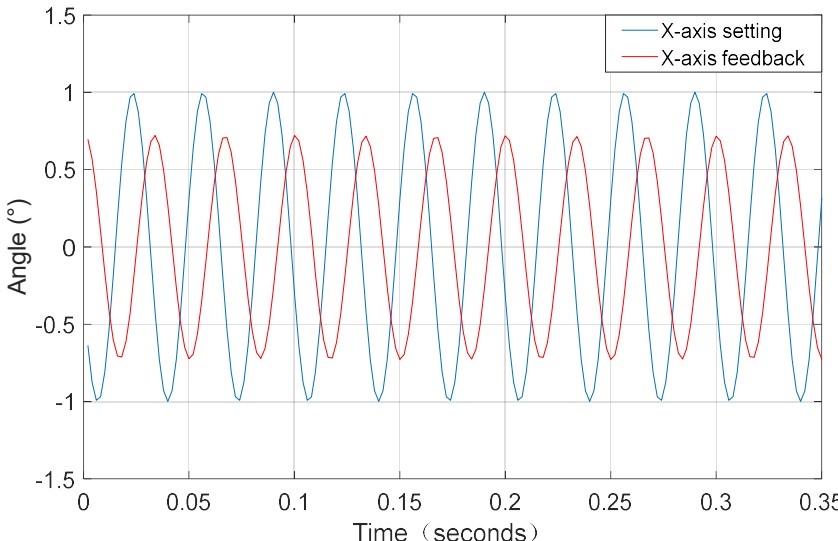

**Figure 9.** Response bandwidth test after decoupling given a 300 Hz sinusoidal on the X-axis.

In addition, a phase difference test of FSM was carried out by using a sinusoidal curve with a 200 Hz frequency, an amplitude of 1.0 mrad, and a phase difference of 180° between the X-axis and Y-axis. In Figure 10, the position feedback waveform collected by feedback at 10 kHz is shown. The test waveform shows that the phase difference of X/Y-axis has no effect on the movement of each degree of freedom of the FSM.

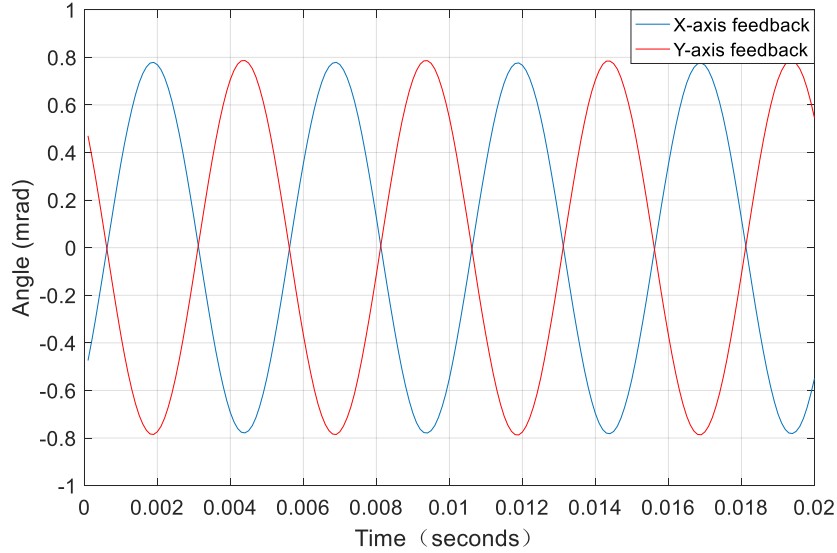

**Figure 10.** Phase difference test of 180 degrees of XY-axis double sinusoidal.

Finally, we tested the antidisturbance performance of the FSM by superimposing 3% of random noise on the 200 Hz sinusoidal input curve with an amplitude of 1mrad. Taking the X-axis as an example, the test curve is shown in Figure 11. It can be seen from the figure that the FSM does not receive the influence of disturbance and has a good dynamic response.

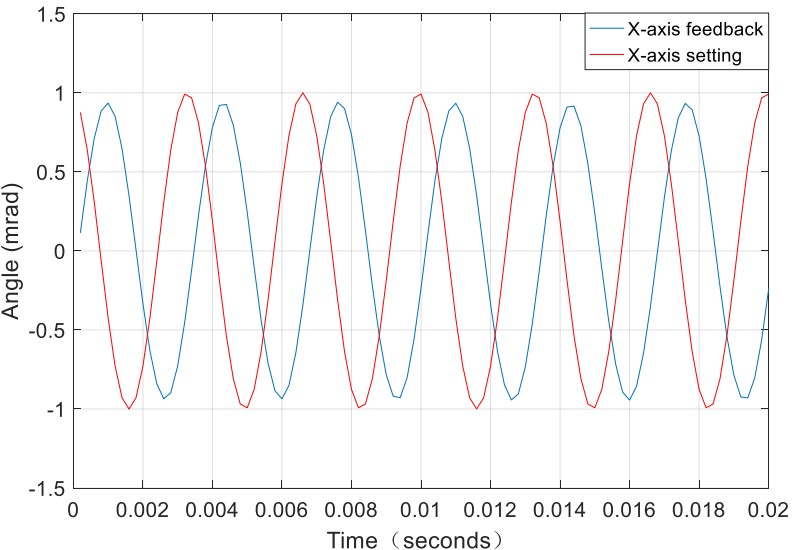

**Figure 11.** X-axis antidisturbance capability test.

### 5.3. Tracking Accuracy Test of 3D GISC LiDAR Tracking UAV (Unmanned Aerial Vehicle)

For the field-tracking experiment test of 3D GISC LiDAR system, the UAV's target range imaging system was 2.74 km, and the maximum flight speed of the UAV was 7 m/s. The comparison of the dual-FSM composite axis tracking accuracy based on the 3D GISC LiDAR system before and after digital FSM decoupling and resonance suppression control is shown in Figure 12a,b, respectively. The pixel angle resolution was 5.1 μrad. Therefore, under the control of the existing algorithm, the azimuth tracking accuracy was 8.3 μrad (σ), the pitch angle tracking accuracy was 15.7 μrad (σ). The tracking accuracy under the control of the algorithm proposed in this paper is shown in Figure 12b, with an azimuth tracking accuracy of 2.2 μrad (σ) and a pitch angle tracking accuracy of 1.5 μrad (σ), which meets the resolution requirements of 3D GISC imaging.

### 5.4. Three-Dimensional GISC LiDAR Tracking UAV Imaging Test

In order to compare the influence of the change of control accuracy before and after dual-FSM decoupling and resonance suppression on imaging, a comparative imaging test of tracking the UAV was carried out for the dual-FSM 3D GISC LiDAR moving-target-tracking imaging system after FSM decoupling and resonance suppression. Figures 13 and 14, respectively, show the test images of the dual-FSM 3D GISC LiDAR contrast imaging for the UAV field tracking before and after FSM decoupling and resonance suppression. The test distance was 2.74 KM, the model of UAV was a DJI phantom 4, and the size was 289.5 × 289.5 × 289.5 mm. During the test, the flight speed of the UAV was (hovering, 2 m/s, and 7 m/s). As shown in Figure 13 below, when the UAV hovers, that is, the relative motion speed of the target is low, the imaging effect of the system is relatively clear. When the moving speed of the target increases, the imaging effect becomes obviously blurred, which is caused by the low tracking accuracy and response speed of the system. Comparing the experimental results of the tracking imaging system after FSM decoupling, as shown in Figure 14 below, when the UAV hovers, the imaging effect of the system is clearer because the accuracy of the tracking imaging system after FSM decoupling is much better than the moving-target imaging radar system before decoupling. Moreover, when the target's moving speed increases, the imaging effect does not change significantly. This result verifies the design and principle of the dual-FSM 3D GISC LiDAR tracking imaging system proposed in this paper.

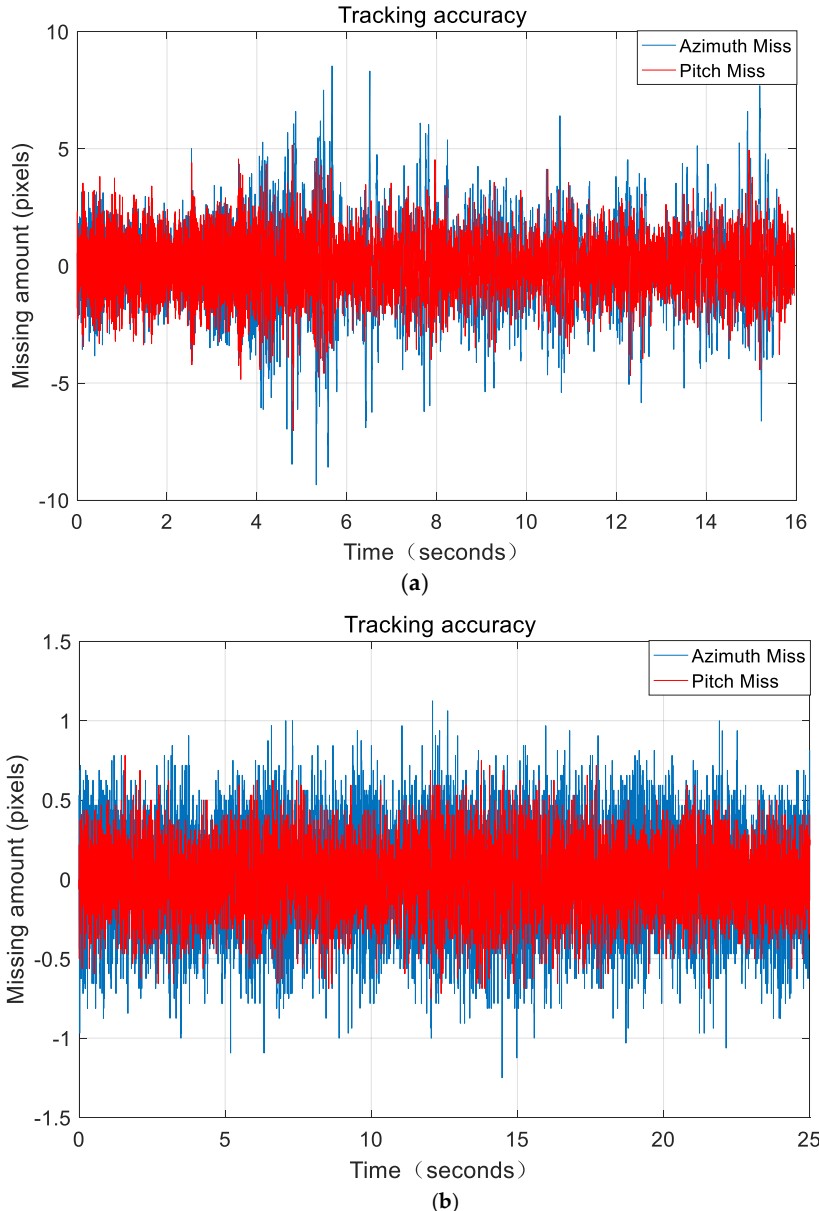

**Figure 12.** Comparative test of tracking accuracy of the azimuth and pitch axis of a UAV in flight. (**a**) FSM is controlled by existing algorithm; (**b**) FSM is controlled by digital decoupling and resonance suppression.

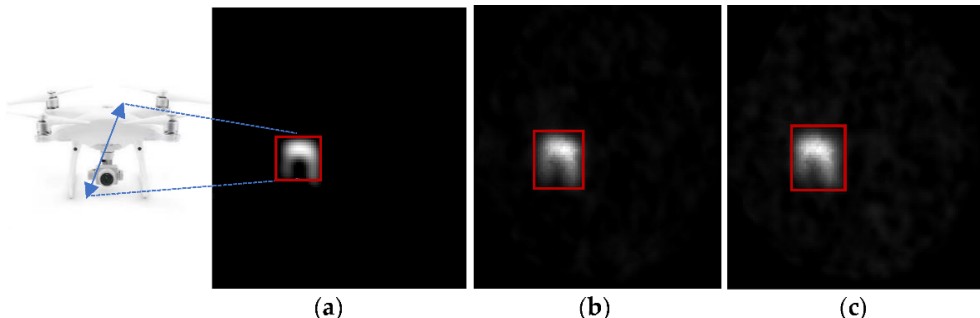

**Figure 13.** Comparison of imaging effects of a dual-FSM 3D GISC LiDAR system with existing control algorithms at different speeds of a UAV (**a**) UAV hovering; (**b**) UAV V = 2 m/s; (**c**) UAV V = 7 m/s.

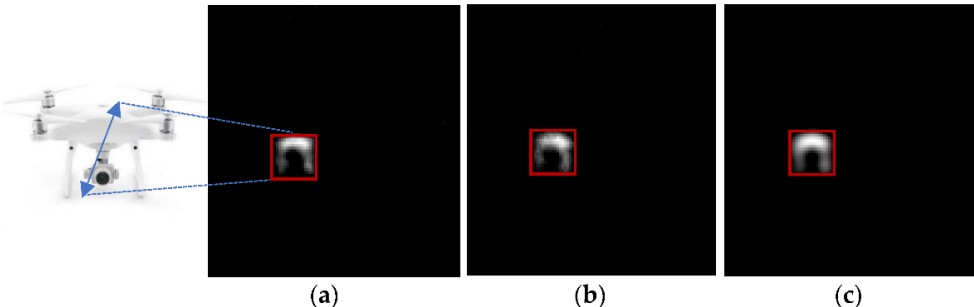

**Figure 14.** Comparison of imaging effects of a dual-FSM 3D GISC LiDAR system under the control of digital decoupling and resonance suppression at different Speeds of a UAV (**a**) UAV hovering; (**b**) UAV V = 2 m/s; (**c**) UAV V = 7 m/s.

## 6. Conclusions

Aiming at the difficulties encountered in the practical application transformation of 3D GISC LiDAR technology, this paper studied a dual-FSM composite axis control method based on 3D GISC LiDAR, and studied the high-resolution imaging detection of high-speed moving targets from three aspects. Firstly, aiming at the problem of noncoaxial sensor installation in digital FSM control, an innovative decoupling method of sensor decoupling and motion decoupling was proposed; this method successfully reduced the coupling from 7% to 0.6%. Secondly, in order to take into account the real-time and high-precision digital FSM control, the control strategy of a position loop and a second-order digital notch filter was proposed to suppress the first-order resonance of the mechanism; the fine tracking bandwidth reached 300 Hz. Thirdly, aiming at the problem of receiving the aperture constraint of the system, this paper presented an optical path design diagram of a dual-FSM 3D GISC LiDAR tracking imaging system. Simulation and experimental results showed the feasibility of the control strategy. Finally, a dual-FSM 3D GISC LiDAR tracking imaging system was developed, and the tracking accuracy reached 2.2 μrad (σ). Based on the images obtained in the test, the high-resolution 3D imaging capability of the LiDAR was first demonstrated. This provides a basis for the further transformation of this research into a practical application of remote sensing imaging detection. At present, the target distance verified by this method is 2.74 km. However, with the further target detection demand of future applications, it is also necessary to study the target position's accurate detection method combined with the echo signal's energy in 3D GISC LiDAR, so that the 3D GISC LiDAR system can play a greater role in the field of target detection.

**Author Contributions:** Conceptualization, X.S. and X.Q.; methodology, Y.C.; software, Y.C.; validation, Y.C., H.W., M.C. and C.W.; formal analysis, Y.C.; resources, W.H.; data curation, X.F.; writing—original draft preparation, Y.C.; writing—review and editing, Y.C., X.Q., M.C. and C.W.; visualization, X.F.; supervision, J.H.; project administration, M.X.; funding acquisition, W.H. All authors have read and agreed to the published version of the manuscript.

**Funding:** This research received no external funding.

**Acknowledgments:** The authors express sincere thanks for the experiments provided by the Photoelectric Tracking and Measurement Technology Laboratory, Xi'an Institute of Optics and Precision Mechanics, CAS and the dual-FSM compound axis tracking and pointing project of the Xi'an Institute of Optics and Precision Mechanics, (grant no. Y99031C2), China.

**Conflicts of Interest:** The authors declare no conflict of interest.

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
