# Peer review of "A Tracking Imaging Control Method for Dual-FSM 3D GISC LiDAR"

_remotesensing, doi:10.3390/rs14133167_

Round 1

Reviewer 1 Report

The authors have revised their manuscript and have responded to the comments. I think this paper can be accepted after some improvements and careful check. My main comments are discussed below.

1The format of the full name should be unified and standardized.

2、“KMin line 29 should be“km”; “Target”in line 49 should be“target”; “methods”in line 203 should be“Methods”; “44hz”in line 250 and line 255 should be“44Hz”;”Urad” in FIGURE 12 should be“μrad”

3What is the meaning of “Sa” and “Sb” in equations (5) and (6)?

4、“Figure 5 in line 229 presents before figure 3 and figure 4.

Author Response

Please see the attachment PDF for the detailed response file.

Reviewer 2 Report

The changes to the manuscript have improved it greatly.  Thank you for your efforts!

Author Response

Deer Reviewer 2:

Thank you very much for your attention and the valuable comments on our paper.

Reviewer 3 Report

The paper addresses an interesting topic.

Too bad the authors sent a version that contains comments from one of them. This negligence is unpleasant for review. Also, regarding the negligent work of the authors, we can mention the numbering and writing of the chapters, the way of writing the information related to the figures or the source of information of some figures (for example figure 1.)

The authors describe in Chapter 4 the design method. In Chapter 5, the authors present the results of the analysis. The way the presentation is made does not explain whether the comparison made by the authors is based on results obtained only from the experimental platform or not. If the data used comes from elsewhere, it is necessary to specify where. If only the data obtained from the experimental platform are used, then there is a problem with the validation of the results obtained.

At the end of the paper, the authors conclude favorably the results obtained. These conclusions are not criticized by the authors, they are considered to be obtained in a study that has no limitations. Correct? Is the method so good?

The study of the authors is one with good potential, but the way they chose to present their work is deficient. The paper needs to be improved a lot.

Author Response

Dear Reviewer 3:

We would like to give our special thanks to you for providing valuable comments and suggestions for the paper. We revise the paper accordingly. Please see the attachment PDF for the detailed response file.

Reviewer 4 Report

After a careful review of the manuscript, I believe the quality of the work is improved in comparison with the previously-submitted version. Generally speaking, the level of contributions and the science behind the work are fairly acceptable now, however, the presentation and the language still need further effort to be improved.

Author Response

Dear Reviewer 4:

We would like to give our special thanks to you for providing valuable comments and suggestions for the paper. We have modified the language and presentation in the article and reviewed the whole manuscript carefully.

Round 2

Reviewer 3 Report

The improvements made by the authors clarify the weaknesses we have noticed.

Usually, in such a paper, at the end of it, the authors make clarifications about the contribution of each author to the research. I recommend the authors to complete these details in accordance with the requirements of the journal.

This manuscript is a resubmission of an earlier submission. The following is a list of the peer review reports and author responses from that submission.

Round 1

Reviewer 1 Report

The paper addresses the problem of target tracking using the n 3D GISC LiDAR. At the center of the work is the concept of Fast Steering Mirror and the mechanical problems that it encounters. I must admit that I am not an expert in the mechanical and electromechanical aspects addressed by the authors, and therefore I cannot judge whether the improvements in regard to mechanical coupling are worthwile or not.

I was expecting to see more details about the tracking process which ensures stable results in the presence of a moving target. There are some schematic details, but no algorithm for tracking is presented, although the title and the abstract lead me to believe that this was the main contribution of the paper. It seems that the tracking algorithm was proposed by "s, the team of Shanghai Institute of Optics and 299
Fine Mechanics Chinese Academy of Sciences proposed a moving target imaging radar system based on single pendulum mirror video tracking[4]", and the current work is merely an improvement. I would suggest that the authors present the original algorithm as well, at least briefly, and also show some differential results to show the effect of the tracking vs no tracking on the final results.

There are some mistakes, such as this paragraph that was from the template:

"Research manuscripts reporting large datasets that are deposited in a publicly avail-able database should specify where the data have been deposited and provide the relevant accession numbers. If the accession numbers have not yet been obtained at the time of submission, please state that they will be provided during review. They must be provided prior to publication."

Reviewer 2 Report

Lines 191 to 195 seem to be a leftover template that should be removed.

Line 222 figure should be figure 5, not 6.

Lines 233 to 238 should be edited to remove redundant sentances.

Figure 6 is mislabelled as figure 7 at line 322

At line 357, the Figure referenced should be Figure 7, noy Figure 8

Reviewer 3 Report

The main idea of the manuscript is quite clear and the explanation is well presented. 

However, the flow is not kind enough to clearly understand the contributions of the authors, so the following question arises.

1. In the introduction, it was said that dual FSM GISC was proposed to overcome image resolution degradation. In this regard, what is the key difference between the existing method and the proposed dual FSM method? Can you objectively compare and analyze the results of the difference with the results of the existing method?

2. In terms of tracking and aiming control, what exactly is the contribution of authors that differentiates the proposed one from existing methods?
